# ROBUST CONSTRAINED REINFORCEMENT LEARNING

## ABSTRACT

Constrained reinforcement learning is to maximize the reward subject to constraints on utilities/costs. However, in practice it is often the case that the training environment is not the same as the test one, due to, e.g., modeling error, adversarial attack, non-stationarity, resulting in severe performance degradation and more importantly constraint violation in the test environment. To address this challenge, we formulate the framework of robust constrained reinforcement learning under model uncertainty, where the MDP is not fixed but lies in some uncertainty set. The goal is two fold: 1) to guarantee that constraints on utilities/costs are satisfied for all MDPs in the uncertainty set, and 2) to maximize the worst-case reward performance over the uncertainty set. We design a robust primal-dual approach, and further develop theoretical guarantee on its convergence, complexity and robust feasibility. We then investigate a concrete example of $\delta$-contamination uncertainty set, design an online and model-free algorithm and theoretically characterize its sample complexity.

## 1 INTRODUCTION

In many practical reinforcement learning (RL) applications, it is critical for an agent to meet certain constraints on utilities/costs while maximizing the reward. This problem is usually modeled as the constrained Markov decision processes (CMDPs) (Altman, 1999). Consider a CMDP with state space $\mathcal{S}$, action space $\mathcal{A}$, transition kernel $\mathsf{P} = \{p_s^a \in \Delta_\mathcal{S}{}^1 : s \in \mathcal{S}, a \in \mathcal{A}\}$, reward and utility functions: $r, c_i : \mathcal{S} \times \mathcal{A} \to [0, 1]$, $1 \le i \le m$, and discount factor $\gamma$. The goal of CMDP is to find a stationary policy $\pi : \mathcal{S} \to \Delta_\mathcal{A}$ that maximizes the expected reward subject to constraints on the utility:

$$\max_{\pi \in \Pi} \mathbb{E}_{\pi,\mathsf{P}} \left[ \sum_{t=0}^{\infty} \gamma^t r(S_t, A_t) | S_0 \sim \rho \right], \text{ s.t. } \mathbb{E}_{\pi,\mathsf{P}} \left[ \sum_{t=0}^{\infty} \gamma^t c_i(S_t, A_t) | S_0 \sim \rho \right] \ge b_i, 1 \le i \le m, \quad (1)$$

where $\rho$ is the initial state distribution, $b_i$'s are some thresholds and $\mathbb{E}_{\pi,\mathsf{P}}$ denotes the expectation when the agent follows policy $\pi$ and the environment transits following $\mathsf{P}$.

In practice, it is often the case that the environment on which the learned policy will deploy (the test environment) possibly deviates from the training one, due to, e.g., modeling error of the simulator, adversarial attack, and non-stationarity. This could lead to a significant performance degradation in reward, and more importantly, constraints may not be satisfied anymore, which is severe in safety-critical applications. For example, a drone may run out of battery and crash due to mismatch between training and test environments. This hence motivates the study of robust constrained RL in this paper. In this paper, we take a pessimistic approach in face of uncertainty. Specifically, consider a set of transition kernels $\mathcal{P}$, which is usually constructed in a way to include the test environment with high probability (Iyengar, 2005; Nilim & El Ghaoui, 2004; Bagnell et al., 2001). The learned policy should satisfy the constraints under all these environments in $\mathcal{P}$, i.e., $\forall \mathsf{P} \in \mathcal{P}$,

$$\mathbb{E}_{\pi,\mathsf{P}} \left[ \sum_{t=0}^{\infty} \gamma^t c_i(S_t, A_t) | S_0 \sim \rho \right] \ge b_i, \quad (2)$$

which is equivalent to $\min_{\mathsf{P} \in \mathcal{P}} \mathbb{E}_{\pi,\mathsf{P}} \left[ \sum_{t=0}^{\infty} \gamma^t c(S_t, A_t) | S_0 \sim \rho \right] \ge b_i$. At the same time, we aim to optimize the worst-case reward performance over $\mathcal{P}$:

$$\max_{\pi \in \Pi} \min_{\mathsf{P} \in \mathcal{P}} \mathbb{E}_{\pi,\mathsf{P}} \left[ \sum_{t=0}^{\infty} \gamma^t r(S_t, A_t) | S_0 \sim \rho \right],$$

---

${}^1\Delta_\mathcal{X}$ denotes the probability simplex supported on the set $\mathcal{X}$.

$$\text{s.t. } \min_{\mathsf{P} \in \mathcal{P}} \mathbb{E}_{\pi,\mathsf{P}} \left[ \sum_{t=0}^{\infty} \gamma^t c_i(S_t, A_t) | S_0 \sim \rho \right] \geq b_i, 1 \leq i \leq m. \tag{3}$$

On one hand, a feasible solution to eq. (3) always satisfies eq. (2), and on the other hand, the solution to eq. (3) provides a performance guarantee for any $\mathsf{P} \in \mathcal{P}$. We note that our approach and analysis can also be applied to the optimistic approach in face of uncertainty.

In this paper, we design and analyze a robust primal-dual algorithm for the problem of robust constrained RL. In particular, the technical challenges and our major contributions are as follows.

- We take the Lagrange multiplier method to solve the constrained policy optimization problem. A first question is that whether the primal problem is equivalent to the dual problem, i.e., whether the duality gap is zero. For non-robust constrained RL, the Lagrange function has a zero duality gap (Paternain et al., 2019; Altman, 1999). However, we show that this is not necessarily true in the robust constrained setting. Note that the set of visitation distribution being convex is one key property to show zero duality gap of constrained MDP (Altman, 1999; Paternain et al., 2019). In this paper, we constructed a novel counter example showing that the set of robust visitation distributions for our robust problem is non-convex.

- In the dual problem of non-robust CMDPs, the sum of two value functions is actually a value function of the combined reward. However, this does not hold in the robust setting, since the worst-case transition kernels for the two robust value functions are not necessarily the same. Therefore, the geometry of our Lagrangian function is much more complicated. In this paper, we formulate the dual problem of the robust constrained RL problem as a minimax linear-nonconcave optimization problem, and show that the optimal dual variable is bounded. We then construct a robust primal-dual algorithm by alternatively updating the primal and dual variables. We theoretically prove the convergence to stationary points, and characterize its complexity.

- In general, convergence to stationary points of the Lagrangian function does not necessarily imply that the solution is feasible (Lin et al., 2020; Xu et al., 2020). We design a novel proof to show that the gradient belongs to the normal cone of the feasible set, based on which we further prove the robust feasibility of the obtained policy.

- We apply and extend our results on an important uncertainty set referred to as $\delta$-contamination model (Huber, 1965). Under this model, the robust value functions are not differentiable and we hence propose a smoothed approximation of the robust value function towards a better geometry. We further investigate the practical online and model-free setting and design an actor-critic type algorithm. We also establish its convergence, sample complexity, and robust feasibility.

We then discuss works related to robust constrained RL.

**Robust constrained RL.** In (Russel et al., 2020), the robust constrained RL problem was studied, and a heuristic approach was developed. The basic idea is to estimate the robust value functions, and then to use the vanilla policy gradient method (Sutton et al., 1999) with the vanilla value function replaced by the robust value function. However, this approach did not take into consideration the fact that the worst-case transition kernel is also a function of the policy (see Section 3.1 in (Russel et al., 2020)), and therefore the "gradient" therein is not actually the gradient of the robust value function. Thus, its performance and convergence cannot be theoretically guaranteed. The other work (Mankowitz et al., 2020) studied the same robust constrained RL problem under the continuous control setting, and proposed a similar heuristic algorithm. They first proposed a robust Bellman operator and used it to estimate the robust value function, which is further combined with some non-robust continuous control algorithm to update the policy. Both approaches in (Russel et al., 2020) and (Mankowitz et al., 2020) inherit the heuristic structure of "robust policy evaluation" + "non-robust vanilla policy improvement", which may not necessarily guarantee an improved policy in general. In this paper, we employ a "robust policy evaluation" + "***robust*** policy improvement" approach, which guarantees an improvement in the policy, and more importantly, we provide theoretical convergence guarantee, robust feasibility guarantee, and complexity analysis for our algorithms.

**Constrained RL.** The most commonly used method for constrained RL is the primal-dual method (Altman, 1999; Paternain et al., 2019; 2022; Liang et al., 2018; Stooke et al., 2020; Tessler et al., 2018; Yu et al., 2019; Zheng & Ratliff, 2020; Efroni et al., 2020; Auer et al., 2008), which augments the objective with a sum of constraints weighted by their corresponding Lagrange multipliers, and then alternatively updates the primal and dual variables. It was shown that the strong duality holds

for constrained RL, and hence the primal-dual method has zero duality gap (Paternain et al., 2019; Altman, 1999). The convergence rate of the primal-dual method was investigated in (Ding et al., 2020; 2021; Li et al., 2021b; Liu et al., 2021; Ying et al., 2021). Another class of method is the primal method, which is to enforce the constraints without resorting to the Lagrangian formulation (Achiam et al., 2017; Liu et al., 2020; Chow et al., 2018; Dalal et al., 2018; Xu et al., 2021; Yang et al., 2020). The above studies, when directly applied to *robust* constrained RL, cannot guarantee the constraints when there is model deviation. Moreover, the objective and constraints in this paper take min over the uncertainty set (see eq. (4)), and therefore have much more complicated geometry than the non-robust case.

**Robust RL under model uncertainty.** Model-based robust RL was firstly introduced and studied in (Iyengar, 2005; Nilim & El Ghaoui, 2004; Bagnell et al., 2001; Satia & Lave Jr, 1973; Wiesemann et al., 2013; Lim & Autef, 2019; Xu & Mannor, 2010; Yu & Xu, 2015; Lim et al., 2013; Tamar et al., 2014), where the uncertainty set is assumed to be known, and the problem can be solved using robust dynamic programming. It was then extended to the model-free setting, where the uncertainty set is unknown, and only samples from its centroid can be collected (Roy et al., 2017; Wang & Zou, 2021; 2022; Zhou et al., 2021; Yang et al., 2021; Panaganti & Kalathil, 2021; Ho et al., 2018; 2021). There are also empirical studies on robust RL, e.g., (Vinitsky et al., 2020; Pinto et al., 2017; Abdullah et al., 2019; Hou et al., 2020; Rajeswaran et al., 2017; Huang et al., 2017; Kos & Song, 2017; Lin et al., 2017; Pattanaik et al., 2018; Mandlekar et al., 2017). These works focus on robust RL without constraints, whereas in this paper we investigate robust RL with constraints, which is more challenging. There is a related line of works on (robust) imitation learning (Ho & Ermon, 2016; Fu et al., 2017; Torabi et al., 2018; Viano et al., 2022), which can be formulated as a constrained problem. But their problem settings and approaches are fundamentally different from ours.

## 2 PRELIMINARIES

**Constrained MDP.** Consider the CMDP problem in eq. (1). Define the visitation distribution induced by policy $\pi$ and transition kernel P: $d^\pi_{\rho,\mathsf{P}}(s,a) = (1-\gamma)\sum_{t=0}^\infty \gamma^t \mathbb{P}(S_t = s, A_t = a | S_0 \sim \rho, \pi, \mathsf{P})$. It can be shown that the set of the visitation distributions of all policies $\{d^\pi_{\rho,\mathsf{P}} \in \Delta_{\mathcal{S}\times\mathcal{A}} : \pi \in \Pi\}$ is convex (Paternain et al., 2022; Altman, 1999). Based on this convexity, the strong duality of CMDP can be established (Altman, 1999; Paternain et al., 2019) under a standard assumption referred as Slater's condition: (Bertsekas, 2014; Ding et al., 2021): there exists a constant $\zeta > 0$ and a policy $\pi \in \Pi$ s.t. $\forall i, V^\pi_{c_i,\mathsf{P}} - b_i \geq \zeta$.

**Robust MDP.** In this paper, we focus on the $(s,a)$-rectangular uncertainty set (Nilim & El Ghaoui, 2004; Iyengar, 2005), i.e., $\mathcal{P} = \bigotimes_{s,a} \mathcal{P}^a_s$, where $\mathcal{P}^a_s \subseteq \Delta_{\mathcal{S}}$. At each time step, the environment transits following a transition kernel belonging to the uncertainty set $\mathsf{P}_t \in \mathcal{P}$. The robust value function of a policy $\pi$ is then defined as the worst-case expected accumulative discounted reward following policy $\pi$ over all MDPs in the uncertainty set (Nilim & El Ghaoui, 2004; Iyengar, 2005):

$$V^\pi_{r,\mathcal{P}}(s) \triangleq \min_{\kappa = (\mathsf{P}_0,\mathsf{P}_1,\dots) \in \bigotimes_{t \geq 0} \mathcal{P}} \mathbb{E}_\kappa \left[ \sum_{t=0}^\infty \gamma^t r(S_t, A_t) | S_0 = s, \pi \right], \tag{4}$$

where $\mathbb{E}_\kappa$ denotes the expectation when the state transits according to $\kappa$. It was shown that the robust value function is the fixed point of the robust Bellman operator (Nilim & El Ghaoui, 2004; Iyengar, 2005; Puterman, 2014): $\mathbf{T}_\pi V(s) \triangleq \sum_{a \in \mathcal{A}} \pi(a|s) \left( r(s,a) + \gamma \sigma_{\mathcal{P}^a_s}(V) \right)$, where $\sigma_{\mathcal{P}^a_s}(V) \triangleq \min_{p \in \mathcal{P}^a_s} p^\top V$ is the support function of $V$ on $\mathcal{P}^a_s$.

Note that the minimizer of eq. (4), $\kappa^*$, is stationary in time (Iyengar, 2005), which we denote by $\kappa^* = \{\mathsf{P}^\pi, \mathsf{P}^\pi, \dots\}$, and refer to $\mathsf{P}^\pi$ as the worst-case transition kernel. Then the robust value function $V^\pi_{r,\mathcal{P}}$ is actually the value function under policy $\pi$ and transition kernel $\mathsf{P}^\pi$. The goal of robust RL is to find the optimal robust policy $\pi^*$ that maximizes the worst-case accumulative discounted reward: $\pi^* = \arg\max_\pi V^\pi_{r,\mathcal{P}}(s), \forall s \in \mathcal{S}$.

## 3 ROBUST CONSTRAINED RL

Recall the robust constrained RL formulated in eq. (3):

$$\max_{\theta \in \Theta} V^{\pi_\theta}_r(\rho), \text{ s.t. } V^{\pi_\theta}_{c_i}(\rho) \geq b_i, 1 \leq i \leq m, \tag{5}$$

where for simplicity we omit the subscript $\mathcal{P}$ in $V_{\diamond,\mathcal{P}}^{\pi_\theta}$ and denote by $V_{c_i}^{\pi_\theta}(\rho)$ and $V_r^{\pi_\theta}(\rho)$ the robust value function for $c_i$ and $r$ under $\pi_\theta$. The goal of eq. (5) is to find a policy that maximizes the robust reward value function among those feasible solutions. Here, any feasible solution to eq. (5) can guarantee that under any MDP in the uncertainty set, its accumulative discounted utility is always no less than $b_i$, which guarantees robustness to constraint violation under model uncertainty. Furthermore, the optimal solution to eq. (5) achieves the best "worst-case reward performance" among all feasible solutions. If we use the optimal solution to eq. (5), then under any MDP in the uncertainty set, we have a guaranteed reward no less than the value of eq. (5).

In this paper, we focus on the parameterized policy class, i.e., $\pi_\theta \in \Pi_\Theta$, where $\Theta \subseteq \mathbb{R}^d$ is a parameter set and $\Pi_\Theta$ is a class of parameterized policies, e.g., direct parameterized policy, softmax or neural network policy. For technical convenience, we adopt a standard assumption on the policy class.

**Assumption 1.** *The policy class $\Pi_\Theta$ is $k$-Lipschitz and $l$-smooth, i.e., for any $s \in \mathcal{S}$ and $a \in \mathcal{A}$ and for any $\theta \in \Theta$, there exist universal constants $k, l$, such that $\|\nabla \pi_\theta(a|s)\| \leq k$, and $\|\nabla^2 \pi_\theta(a|s)\| \leq l$.*

This assumption can be satisfied by many policy classes, e.g., direct parameterization (Agarwal et al., 2021), soft-max (Mei et al., 2020; Li et al., 2021a; Wang & Zou, 2020), or neural network with Lipschitz and smooth activation functions (Du et al., 2019; Neyshabur, 2017; Miyato et al., 2018).

The problem eq. (5) is equivalent to the following max-min problem:

$$\max_{\theta \in \Theta} \min_{\lambda_i \geq 0} V_r^{\pi_\theta}(\rho) + \sum_{i=1}^m \lambda_i (V_{c_i}^{\pi_\theta}(\rho) - b_i). \tag{6}$$

Unlike non-robust CMDP, strong duality for robust constrained RL may not hold. For robust RL, the robust value function can be viewed as the value function for policy $\pi$ under its worst-case transition kernel $\mathsf{P}^\pi$, and therefore can be written as the inner product between the reward (utility) function and the visitation distribution induced by $\pi$ and $\mathsf{P}^\pi$ (referred to as robust visitation distribution of $\pi$). The following lemma shows that the set of robust visitation distributions may not be convex, and therefore, the approach used in (Altman, 1999; Paternain et al., 2019) to show strong duality cannot be applied here.

**Lemma 1.** *There exists a robust MDP, such that the set of robust visitation distributions is non-convex.*

In the following, we focus on the dual problem of eq. (6). For simplicity, we investigate the case with one constraint, and extension to the case with multiple constraints is straightforward:

$$\min_{\lambda \geq 0} \max_{\theta \in \Theta} V_r^{\pi_\theta}(\rho) + \lambda (V_c^{\pi_\theta}(\rho) - b). \tag{7}$$

We make an assumption of Slater's condition, assuming there exists at least one strictly feasible policy (Bertsekas, 2014; Ding et al., 2021), under which, we further show that the optimal dual variable of eq. (7) is bounded. This assumption is

**Assumption 2.** *There exists $\zeta > 0$ and a policy $\pi \in \Pi_\Theta$, s.t. $V_c^\pi(\rho) - b \geq \zeta$.*

**Lemma 2.** *Denote the optimal solution of eq. (7) by $(\lambda^*, \pi_{\theta^*})$. Then, $\lambda^* \in \left[0, \frac{2}{\zeta(1-\gamma)}\right]$.*

Lemma 2 suggests that the dual problem eq. (7) is equivalent to a bounded min-max problem:

$$\min_{\lambda \in \left[0, \frac{2}{\zeta(1-\gamma)}\right]} \max_{\theta \in \Theta} V_r^{\pi_\theta}(\rho) + \lambda (V_c^{\pi_\theta}(\rho) - b). \tag{8}$$

The problem in eq. (8) is a bounded linear-nonconcave optimization problem. We then propose our robust primal-dual algorithm for robust constrained RL in Algorithm 1. The basic idea of Algorithm 1 is to perform gradient descent-ascent w.r.t. $\lambda$ and $\theta$ alternatively. When the policy $\pi$ violates the constraint, the dual variable $\lambda$ increases such that $\lambda V_c^\pi$ dominates $V_r^\pi$. Then the gradient ascent will update $\theta$ until the policy satisfies the constraint. Therefore, this approach is expected to find a feasible policy (as will be shown in Lemma 5). Here, $\prod_{\mathcal{X}}(x)$ denotes the projection of $x$ to the set $\mathcal{X}$, and $\{b_t\}$ is a non-negative monotone decreasing sequence, which will be specified later. Algorithm 1 reduces to the vanilla gradient descent-ascent algorithm in (Lin et al., 2020) if $b_t = 0$. However, $b_t$ is critical to the convergence of Algorithm 1 (Xu et al., 2020). The outer problem of eq. (8) is

---

**Algorithm 1** Robust Primal-Dual algorithm (RPD)

---

**Input**: $T$, $\alpha_t$, $\beta_t$, $b_t$
**Initialization**: $\lambda_0$, $\theta_0$
   **for** $t = 0, 1, ..., T-1$ **do**
     $\lambda_{t+1} \leftarrow \prod_{[0,\Lambda^*]} \left( \lambda_t - \frac{1}{\beta_t} \left( V_c^{\pi_{\theta_t}}(\rho) - b \right) - \frac{b_t}{\beta_t} \lambda_t \right)$
     $\theta_{t+1} \leftarrow \prod_{\Theta} \left( \theta_t + \frac{1}{\alpha_t} \left( \nabla_\theta V_r^{\pi_{\theta_t}}(\rho) + \lambda_{t+1} \nabla_\theta V_c^{\pi_{\theta_t}}(\rho) \right) \right)$
   **end for**
**Output**: $\theta_T$

---

actually linear, and after introducing $b_t$, the update of $\lambda_t$ can be viewed as a gradient descent of a strongly-convex function $\lambda(V_c - b) + \frac{b_t}{2}\lambda^2$, which converges more stable and faster.

Denote that Lagrangian function by $V^L(\theta, \lambda) \triangleq V_r^{\pi_\theta}(\rho) + \lambda(V_c^{\pi_\theta}(\rho) - b)$, and further denote the gradient mapping of Algorithm 1 by

$$G_t \triangleq \left[ \begin{array}{c} \beta_t \left( \lambda_t - \prod_{[0,\Lambda^*]} \left( \lambda_t - \frac{1}{\beta_t} \left( \nabla_\lambda V^L(\theta_t, \lambda_t) \right) \right) \right) \\ \alpha_t \left( \theta_t - \prod_{\Theta} \left( \theta_t + \frac{1}{\alpha_t} \left( \nabla_\theta V^L(\theta_t, \lambda_t) \right) \right) \right) \end{array} \right]. \tag{9}$$

The gradient mapping is a standard measure of convergence for projected optimization approaches (Beck, 2017). Intuitively, it reduces to the gradient $(\nabla_\lambda V^L, \nabla_\theta V^L)$, when $\Lambda^* = \infty$ and $\Theta = \mathbb{R}^d$, and it measures the updates of $\theta$ and $\lambda$ at time step $t$. If $\|G_t\| \to 0$, the updates of both variables are small, and hence the algorithm converges to a stationary solution.

To show the convergence of Algorithm 1, we make the following Lipschitz smoothness assumption.

**Assumption 3.** *The gradients of the Lagrangian function are Lipschitz:*

$$\|\nabla_\lambda V^L(\theta, \lambda)|_{\theta_1} - \nabla_\lambda V^L(\theta, \lambda)|_{\theta_2}\| \le L_{11}\|\theta_1 - \theta_2\|, \tag{10}$$

$$\|\nabla_\lambda V^L(\theta, \lambda)|_{\lambda_1} - \nabla_\lambda V^L(\theta, \lambda)|_{\lambda_2}\| \le L_{12}|\lambda_1 - \lambda_2|, \tag{11}$$

$$\|\nabla_\theta V^L(\theta, \lambda)|_{\theta_1} - \nabla_\theta V^L(\theta, \lambda)|_{\theta_2}\| \le L_{21}\|\theta_1 - \theta_2\|, \tag{12}$$

$$\|\nabla_\theta V^L(\theta, \lambda)|_{\lambda_1} - \nabla_\theta V^L(\theta, \lambda)|_{\lambda_2}\| \le L_{22}|\lambda_1 - \lambda_2|. \tag{13}$$

As will be shown in Section 4, Assumption 3 can be satisfied with a smoothed approximation of the robust value function.

In the following theorem, we show that our robust primal-dual algorithm converges to a stationary point of the min-max problem eq. (16), with a complexity of $\mathcal{O}(\epsilon^{-4})$.

**Theorem 1.** *Under Assumption 3, if we set step sizes $\alpha_t, \beta_t$, and $b_t$ as in Section J and $T = \mathcal{O}(\epsilon^{-4})$, then $\min_{1 \le t \le T} \|G_t\| \le 2\epsilon$.*

The next proposition characterizes the feasibility of the obtained policy.

**Proposition 1.** *Denote by $W \triangleq \arg\min_{1 \le t \le T} \|G_t\|$. If $\lambda_W - \frac{1}{\beta_W} \left( \nabla_\lambda V_\sigma^L(\theta_W, \lambda_W) \right) \in [0, \Lambda^*)$, then $\pi_W$ satisfies the constraint with a $2\epsilon$-violation.*

In general, convergence to stationary points of the Lagrangian function does not necessarily imply that the solution is feasible. Proposition 1 shows that Algorithm 1 always return a policy that is robust feasible, i.e., satisfying the constraints in eq. (5). Intuitively, if we set $\Lambda^*$ larger so that the optimal solution $\lambda^* \in [0, \Lambda^*)$, then Algorithm 1 is expected to converge to an interior point of $[0, \Lambda^*]$ and therefore, $\pi_W$ is feasible. On the other hand, $\Lambda^*$ can't be set too large. Note that the complexity in Theorem 1 depends on $\Lambda^*$ (see eq. (59) in the appendix), and a larger $\Lambda^*$ means a higher complexity.

## 4 $\delta$-CONTAMINATION UNCERTAINTY SET

In this section, we investigate a concrete example of robust constrained RL with $\delta$-contamination uncertainty set. The method we developed here can be similarly extended to other type of uncertainty

sets like KL-divergence or total variation. The $\delta$-contamination uncertainty set models the scenario where the state transition of the MDP could be arbitrarily perturbed with a small probability $\delta$. This model is widely used to model distributional uncertainty in the literature of robust learning and optimization, e.g., (Huber, 1965; Du et al., 2018; Huber & Ronchetti, 2009; Nishimura & Ozaki, 2004; 2006; Prasad et al., 2020a;b; Wang & Zou, 2021; 2022). Specifically, let $P = \{p_s^a | s \in \mathcal{S}, a \in \mathcal{A}\}$ be the centroid transition kernel, then the $\delta$-contamination uncertainty set centered at P is defined as $\mathcal{P} \triangleq \bigotimes_{s \in \mathcal{S}, a \in \mathcal{A}} \mathcal{P}_s^a$, where $\mathcal{P}_s^a \triangleq \{(1 - \delta)p_s^a + \delta q | q \in \Delta_\mathcal{S}\}, s \in \mathcal{S}, a \in \mathcal{A}$.

Under the $\delta$-contamination setting, the robust Bellman operator can be explicitly computed: $\mathbf{T}_\pi V(s) = \sum_{a \in \mathcal{A}} \pi(a|s) \left(r(s,a) + \gamma \left(\delta \min_{s'} V(s') + (1 - \delta) \sum_{s' \in \mathcal{S}} p_{s,s'}^a V(s')\right)\right)$. In this case, the robust value function is non-differentiable due to the $\min$ term, and hence Assumption 3 does not hold. One possible approach is to use sub-gradient, which, however, is less stable, and its convergence is difficult to characterize. In the following, we design a differentiable and smooth approximation of the robust value function. Specifically, consider a smoothed robust Bellman operator $\mathbf{T}_\sigma^\pi$ using the LSE function:

$$\mathbf{T}_\sigma^\pi V(s) = \mathbb{E}_{A \sim \pi(\cdot|s)} \left[r(s, A) + \gamma(1 - \delta) \sum_{s' \in \mathcal{S}} p_{s,s'}^A V(s') + \gamma \delta \mathrm{LSE}(\sigma, V)\right], \tag{14}$$

where $\mathrm{LSE}(\sigma, V) = \frac{\log(\sum_{i=1}^d e^{\sigma V(i)})}{\sigma}$ for $V \in \mathbb{R}^d$ and some $\sigma < 0$. The approximation error $|\mathrm{LSE}(\sigma, V) - \min V| \to 0$ as $\sigma \to -\infty$, and hence the fixed point of $\mathbf{T}_\sigma^\pi$, denoted by $V_\sigma^\pi$, is an approximation of the robust value function $V^\pi$ (Wang & Zou, 2022). We refer to $V_\sigma^\pi$ as the smoothed robust value function and define the smoothed robust action-value function as $Q_\sigma^\pi(s, a) \triangleq r(s, a) + \gamma(1 - \delta) \sum_{s' \in \mathcal{S}} p_{s,s'}^a V_\sigma^\pi(s') + \gamma \delta \mathrm{LSE}(\sigma, V_\sigma^\pi)$. It can be shown that for any $\pi$, as $\sigma \to -\infty$, $\|V_r^\pi - V_{\sigma,r}^\pi\| \to 0$ and $\|V_c^\pi - V_{\sigma,c}^\pi\| \to 0$.

The gradient of $V_\sigma^{\pi_\theta}$ can be computed explicitly (Wang & Zou, 2022): $\nabla V_\sigma^{\pi_\theta}(s) = B(s, \theta) + \frac{\gamma \delta \sum_{s \in \mathcal{S}} e^{\sigma V_\sigma^{\pi_\theta}(s)} B(s,\theta)}{(1-\gamma) \sum_{s \in \mathcal{S}} e^{\sigma V_\sigma^{\pi_\theta}(s)}}$, where $B(s, \theta) \triangleq \frac{1}{1-\gamma+\gamma\delta} \sum_{s' \in \mathcal{S}} d_{s,\mathsf{P}}^{\pi_\theta}(s') \sum_{a \in \mathcal{A}} \nabla_\theta \pi(a|s') Q_\sigma^{\pi_\theta}(s', a)$, and $d_{s,\mathsf{P}}^{\pi_\theta}(\cdot)$ is the visitation distribution of $\pi_\theta$ under P starting from $s$. Denote the smoothed Lagrangian function by $V_\sigma^L(\theta, \lambda) \triangleq V_{\sigma,r}^{\pi_\theta}(\rho) + \lambda(V_{\sigma,c}^{\pi_\theta}(\rho) - b)$. The following lemma shows that $\nabla V_\sigma^L$ is Lipschitz.

**Lemma 3.** $\nabla V_\sigma^L$ is Lipschitz in $\theta$ and $\lambda$. And hence Assumption 3 holds for $V_\sigma^L$.

A natural idea is to use the smoothed robust value functions to replace the ones in eq. (7):

$$\min_{\lambda \geq 0} \max_{\pi \in \Pi_\Theta} V_{\sigma,r}^\pi(\rho) + \lambda(V_{\sigma,c}^\pi(\rho) - b). \tag{15}$$

As will be shown below in Lemma 6, this approximation can be arbitrarily close to the original problem in eq. (7) as $\sigma \to -\infty$. We first show that under Assumption 2, the following Slater's condition holds for the smoothed problem in eq. (15).

**Lemma 4.** Let $\sigma$ be sufficiently small such that $\|V_{\sigma,c}^\pi - V_c^\pi\| < \zeta$ for any $\pi$, then there exists $\zeta' > 0$ and a policy $\pi' \in \Pi_\Theta$ s.t. $V_{\sigma,c}^{\pi'}(\rho) - b \geq \zeta'$.

The following lemma shows that the optimal dual variable for eq. (15) is also bounded.

**Lemma 5.** Denote the optimal solution of eq. (15) by $(\lambda^*, \pi_{\theta^*})$. Then $\lambda^* \in \left[0, \frac{2C_\sigma}{\zeta'}\right]$, where $C_\sigma$ is the upper bound of smoothed robust value functions $V_{\sigma,c}^\pi$.

Denote by $\Lambda^* = \max\left\{\frac{2C_\sigma}{\zeta'}, \frac{2}{\zeta(1-\gamma)}\right\}$, then problems eq. (8) and eq. (15) are equivalent to the following bounded ones: $\min_{\lambda \in [0,\Lambda^*]} \max_{\pi \in \Pi_\Theta} V_r^\pi(\rho) + \lambda(V_c^\pi(\rho) - b)$, and

$$\min_{\lambda \in [0,\Lambda^*]} \max_{\pi \in \Pi_\Theta} V_{\sigma,r}^\pi(\rho) + \lambda(V_{\sigma,c}^\pi(\rho) - b). \tag{16}$$

The following lemma shows that the two problems are within a gap of $\mathcal{O}(\epsilon)$.

**Lemma 6.** Choose a small enough $\sigma$ such that $\|V_r^\pi - V_{\sigma,r}^\pi\| \leq \epsilon$ and $\|V_c^\pi - V_{\sigma,c}^\pi\| \leq \epsilon$. Then

$$\left| \min_{\lambda \in [0,\Lambda^*]} \max_{\pi \in \Pi_\Theta} V_{\sigma,r}^\pi(\rho) + \lambda(V_{\sigma,c}^\pi(\rho) - b) - \min_{\lambda \in [0,\Lambda^*]} \max_{\pi \in \Pi_\Theta} V_r^\pi(\rho) + \lambda(V_c^\pi(\rho) - b) \right| \leq (1 + \Lambda^*)\epsilon.$$

In the following, we hence focus on the smoothed dual problem in eq. (16), which is an accurate approximation of the original problem eq. (8). Denote the gradient mapping of the smoothed Lagrangian function $V_\sigma^L$ by

$$G_t^\sigma \triangleq \begin{bmatrix} \beta_t \left( \lambda_t - \prod_{[0,\Lambda^*]} \left( \lambda_t - \frac{1}{\beta_t} \left( \nabla_\lambda V_\sigma^L(\theta_t, \lambda_t) \right) \right) \right) \\ \alpha_t \left( \theta_t - \prod_\Theta \left( \theta_t + \frac{1}{\alpha_t} \left( \nabla_\theta V_\sigma^L(\theta_t, \lambda_t) \right) \right) \right) \end{bmatrix}. \tag{17}$$

Applying our RPD algorithm in eq. (16), we have the following convergence guarantee.

**Corollary 1.** *If we set step sizes $\alpha_t, \beta_t$, and $b_t$ as in Section J and set $T = \mathcal{O}(\epsilon^{-4})$, then $\min_{1 \le t \le T} \|G_t^\sigma\| \le 2\epsilon$.*

This corollary implies that our robust primal-dual algorithm converges to a stationary point of the min-max problem eq. (16) under the $\delta$-contamination model, with a complexity of $\mathcal{O}(\epsilon^{-4})$.

---

**Algorithm 2** Smoothed Robust TD (Wang & Zou, 2022)

---

**Input**: $T_{\text{inner}}, \pi, \sigma, c$
**Initialization**: $Q_0, s_0$
    **for** $t = 0, 1, ..., T_{\text{inner}} - 1$ **do**
      Choose $a_t \sim \pi(\cdot|s_t)$ and observe $c_t, s_{t+1}$
      $V_t(s) \leftarrow \sum_{a \in \mathcal{A}} \pi(a|s) Q_t(s, a)$ for all $s \in \mathcal{S}$
      $Q_{t+1}(s_t, a_t) \leftarrow Q_t(s_t, a_t) + \alpha_t \big( c_t + \gamma(1 - \delta) \cdot V_t(s_{t+1}) + \gamma\delta \cdot \text{LSE}(\sigma, V_t) - Q_t(s_t, a_t) \big)$
    **end for**
**Output**: $Q_{T_{\text{inner}}, c} \triangleq Q_{T_{\text{inner}}}$

---

Note that Algorithm 1 assumes knowledge of the smoothed robust value functions which may not be available in practice. Different from the non-robust value function which can be estimated using Monte Carlo, robust value functions are the value function corresponding to the worst-case transition kernel from which no samples are directly taken. To solve this issue, we adopt the smoothed robust TD algorithm (Algorithm 2) from (Wang & Zou, 2022) to estimate the smoothed robust value functions.

It was shown that the smoothed robust TD algorithm converges to the smoothed robust value function with a sample complexity of $\mathcal{O}(\epsilon^{-2})$ (Wang & Zou, 2022) under the tabular case. We then construct our online and model-free RPD algorithm as in Algorithm 3. We note that Algorithm 3 is for the tabular setting with finite $\mathcal{S}$ and $\mathcal{A}$. It can be easily extended to the case with large/continuous $\mathcal{S}$ and $\mathcal{A}$ using function approximation.

---

**Algorithm 3** Online Robust Primal-Dual algorithm

---

**Input**: $T, \sigma, \epsilon_{\text{est}}, \beta_t, \alpha_t, b_t, r, c$
**Initialization**: $\lambda_0, \theta_0$
    **for** $t = 0, 1, ..., T - 1$ **do**
      Set $T_{\text{inner}} = \mathcal{O}\left( \frac{(t+1)^{1.5}}{\epsilon_{\text{est}}^2} \right)$ and run Algorithm 2 for $r$ and $c$, output $Q_{T_{\text{inner}}, r}, Q_{T_{\text{inner}}, c}$
      $\hat{V}_{\sigma,r}^{\pi_{\theta_t}}(s) \leftarrow \sum_a \pi_{\theta_t}(a|s) Q_{T_{\text{inner}}, r}(s, a), \hat{V}_{\sigma,c}^{\pi_{\theta_t}}(s) \leftarrow \sum_a \pi_{\theta_t}(a|s) Q_{T_{\text{inner}}, c}(s, a)$
      $\hat{V}_{\sigma,r}^{\pi_{\theta_t}}(\rho) \leftarrow \sum_s \rho(s) \hat{V}_{\sigma,r}^{\pi_{\theta_t}}(s), \hat{V}_{\sigma,c}^{\pi_{\theta_t}}(\rho) \leftarrow \sum_s \rho(s) \hat{V}_{\sigma,c}^{\pi_{\theta_t}}(s)$
      $\lambda_{t+1} \leftarrow \prod_{[0,\Lambda^*]} \left( \lambda_t - \frac{1}{\beta_t} \left( \hat{V}_{\sigma,c}^{\pi_{\theta_t}}(\rho) - b \right) - \frac{b_t}{\beta_t} \lambda_t \right)$
      $\theta_{t+1} \leftarrow \prod_\Theta \left( \theta_t + \frac{1}{\alpha_t} \left( \nabla_\theta \hat{V}_{\sigma,r}^{\pi_{\theta_t}}(\rho) + \lambda_{t+1} \nabla_\theta \hat{V}_{\sigma,c}^{\pi_{\theta_t}}(\rho) \right) \right)$
    **end for**
**Output**: $\theta_T$

---

Algorithm 3 can be viewed as a biased stochastic gradient descent-ascent algorithm. It is a sample-based algorithm without assuming any knowledge of robust value functions, and can be performed in an online fashion. We further extend the convergence results in Theorem 1 to the model-free setting, and characterize the following finite-time error bound of Algorithm 3. Similarly, Algorithm 3 can be shown to achieve a $2\epsilon$-feasible policy almost surely.

Under the online model-free setting, the estimation of the robust value functions is biased. Therefore, the analysis is more challenging than the existing literature, where it is usually assumed that the gradients are exact. We develop a new method to bound the bias accumulated in every iteration of the algorithm, and establish the final convergence results.

**Theorem 2.** *Consider the same conditions as in Theorem 1. Let $\epsilon_{est} = \mathcal{O}(\epsilon^2)$ and $T = \mathcal{O}(\epsilon^{-4})$, then* $\min_{1 \leq t \leq T} \|G_t^\sigma\| \leq (1 + \sqrt{2})\epsilon$.

## 5 NUMERICAL RESULTS

In this section, we numerically demonstrate the robustness of our algorithm in terms of both maximizing robust reward value function and satisfying constraints under model uncertainty. We compare our RPD algorithm with the heuristic algorithms in (Russel et al., 2021; Mankowitz et al., 2020) and the vanilla non-robust primal-dual method. Based on the idea of "robust policy evaluation" + "non-robust policy improvement" in (Russel et al., 2021; Mankowitz et al., 2020), we combine the robust TD algorithm 2 with non-robust vanilla policy gradient method (Sutton et al., 1999), which we refer to as the heuristic primal-dual algorithm. Several environments, including Garnet (Archibald et al., 1995), $8 \times 8$ Frozen-Lake and Taxi environments from OpenAI (Brockman et al., 2016), are investigated.

We first run the algorithms and store the obtained policies $\pi_t$ at each time step. Then we run the non-smoothed robust TD (Alg 3 in (Wang & Zou, 2022)) with a sample size 200 for 30 times to estimate the non-smoothed objective $V_r(\rho)$ and the non-smoothed constraint $V_c(\rho)$. We then plot them v.s. the number of iterations $t$. The upper and lower envelopes of the curves correspond to the 95 and 5 percentiles of the 30 curves, respectively. We repeat the experiment for two different values of $\delta = 0.2, 0.3$.

**Garnet problem.** A Garnet problem can be specified by $\mathcal{G}(S_n, A_n)$, where the state space $\mathcal{S}$ has $S_n$ states $(s_1, ..., s_{S_n})$ and action space has $A_n$ actions $(a_1, ..., a_{A_n})$. The agent can take any actions in any state, and receives a randomly generated reward/utility signal generated from the uniform distribution on [0,1]. The transition kernels are also randomly generated. The comparison results are shown in Fig.1.

$8 \times 8$ **Frozen-Lake problem.** We then compare the three algorithms under the $8 \times 8$ Frozen-lake problem setting in Fig.2. The Frozen-Lake problem involves a frozen lake of size $8 \times 8$ which contains several "holes". The agent aims to cross the lake from the start point to the end point without falling into any holes. The agent receives $r = -10$ and $c = 0$ when falling in a hole, receives $r = 20$ and $c = 1$ when arrive at the end point; At other times, the agent receives $r = 0$ and a randomly generated utility $c$ according to the uniform distribution on [0,1].

**Taxi problem.** We then compare the three algorithms under the Taxi problem environment. The taxi problem simulates a taxi driver in a $5 \times 5$ map. There are four designated locations in the grid world and a passenger occurs at a random location of the designated four locations at the start of each episode. The goal of the driver is to first pick up the passenger and then to drop off at another specific location. The driver receives $r = 20$ for each successful drop-off, and always receives $r = -1$ at other times. We randomly generate the utility according to the uniform distribution on [0,1] for each state-action pair. The results are shown in Fig.3.

From the experiment results above, it can be seen that: (1) Both our RPD algorithm and the heuristic primal-dual approach find feasible policies satisfying the constraint robustly, i.e., the non-smoothed robust utility functions lie above the threshold $V_c^\pi \geq b$. However, the non-robust primal-dual method fails to find a feasible solution that satisfy the constraint under the worst-case scenario. (2) Compared to the heuristic PD method, our RPD method can obtain more reward and can find a more robust policy while satisfying the robust constraint. Note that the non-robust PD method obtain more reward, but this is because the policy it finds violates the robust constraint. Our experiments demonstrate that among the three algorithms, our RPD algorithm is the best one which optimizes the worst-case reward performance while satisfying the robust constraints on the utility.

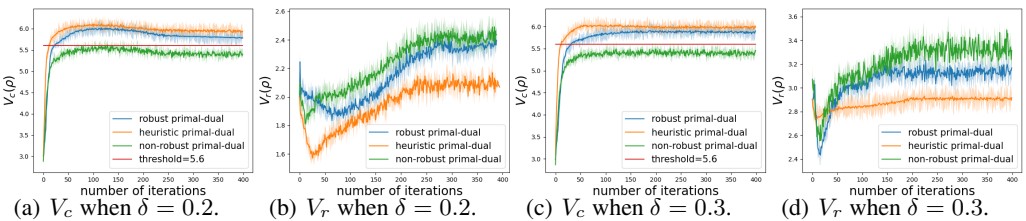

(a) $V_c$ when $\delta = 0.2$.     (b) $V_r$ when $\delta = 0.2$.     (c) $V_c$ when $\delta = 0.3$.     (d) $V_r$ when $\delta = 0.3$.

Figure 1: Comparison on Garnet Problem $\mathcal{G}(20, 10)$.

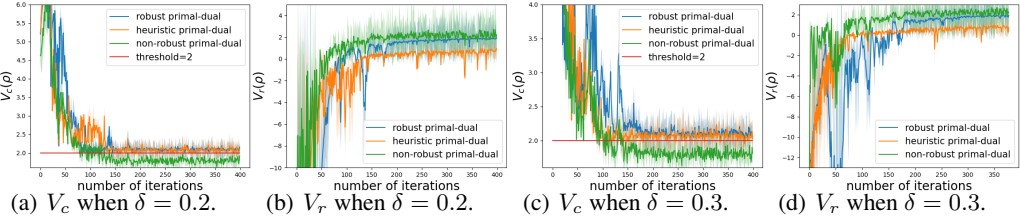

(a) $V_c$ when $\delta = 0.2$.     (b) $V_r$ when $\delta = 0.2$.     (c) $V_c$ when $\delta = 0.3$.     (d) $V_r$ when $\delta = 0.3$.

Figure 2: Comparison on $8 \times 8$ Frozen-Lake Problem.

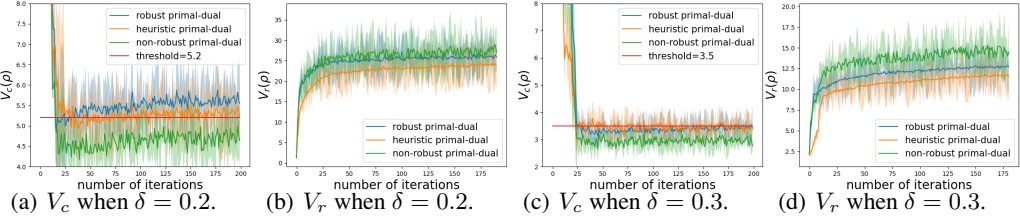

(a) $V_c$ when $\delta = 0.2$.     (b) $V_r$ when $\delta = 0.2$.     (c) $V_c$ when $\delta = 0.3$.     (d) $V_r$ when $\delta = 0.3$.

Figure 3: Comparison on Taxi Problem.

## 6 CONCLUSION

In this paper, we formulate the problem of robust constrained reinforcement learning under model uncertainty, where the goal is to guarantee that constraints are satisfied for all MDPs in the uncertainty set, and to maximize the worst-case reward performance over the uncertainty set. We propose a robust primal-dual algorithm, and theoretically characterize its convergence, complexity and robust feasibility. Our algorithm guarantees convergence to a feasible solution, and outperforms the other two heuristic algorithms. We further investigate a concrete example with $\delta$-contamination uncertainty set, and construct online and model-free robust primal-dual algorithm. Our methodology can also be readily extended to problems with other uncertainty sets like KL-divergence, total variation and Wasserstein distance. The major challenge lies in deriving the robust policy gradient, and further designing model-free algorithm to estimate the robust value function.

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
