# OpenReview forum: " Robust Constrained Reinforcement Learning"
_ICLR.cc/2023/Conference — Submitted to ICLR 2023_

### Official Review · Reviewer_yY2u · 2022-10-24

**Confidence:** 5
**Correctness:** 3
**Technical Novelty And Significance:** 2
**Empirical Novelty And Significance:** 2
**Recommendation:** 3

**Clarity, Quality, Novelty And Reproducibility:**

The novelty of this work seems limited. Moreover, there is some room for improvement in terms of the theoretical results -- stronger guarantees of the learned policy might be possible.

**Strength And Weaknesses:**

Strength:

The model of robust CMDP seems a novel setting that has not been extensively studied in the literature. The algorithm based on Lagrange multiplier is reasonable as it is an adaptation of the existing algorithm for CMDP. (Although I have certain reservations about extending it to the robust setting. See below.)

Weakness:

1. Novelty: The main issue of this work seems a lack of novelty. Both the standard CMDP and robust MDP have been extensively studied in the literature, and this work seems a direct combination of these strands of research. For example, the algorithm is based on the Lagrange multiplier and policy gradient method for CMDP. Seen from the presentation of the algorithm, the authors even omit the dependency on transition $P$, which makes the algorithm extremely similar to a method for CMDP. In terms of analysis, it seems unclear what additional challenge is caused by ensuring robustness with respect to the ambiguity set.

2. Algorithm. It would be great to carefully derive the algorithm. In particular, I think the transformation based on Lagrange multiplier might not be as simple as shown in the paper. In particular, here the functions  $V_r$ and $V_c$ are actually
$$
V_{r} = \inf_{P \in \mathcal{P} } V_{P, r} , \qquad V_{c} = \inf_{P \in \mathcal{P} } V_{P, c} .
$$
When you write down the Lagrangian, do you consider
$$
\inf_{P \in \mathcal{P} } V_{P, r} + \lambda \cdot \inf_{P \in \mathcal{P} } V_{P, c}
$$
or
$$
\inf_{P \in \mathcal{P} } \{  V_{P, r} + \lambda \cdot  V_{P, c} \}
$$

3. Gradient descent ascent. Following the above comment, since the objective involves $\inf_{P \in \mathcal{P} } $, when taking the gradient with respect to the policy parameter and $\lambda$, it seems unclear how the gradient is computed. How do you differentiate through $\inf_P$ and how do you estimate the gradient?

4. Convergence to a stationary point. This paper only shows convergence to a stationary point. However, various existing works on CMDP have established convergence to the globally optimal policy and even ensured zero constraint violation. It would be nice to see if the theoretical guarantees in this work can be improved. Convergence to a stationary point seems not to lead to any optimality certificate of the learned policy.



**Summary Of The Paper:**

This paper studies the constrained robust MDP problem where the goal is to learn a policy that maximizes the expected rewards, subject to the safety constraint that the expected cost exceeds certain thresholds. Moreover, the transition model is chosen from an ambiguity set. As a result, the goal is to maximize the reward under the most adversarial model, while ensuring that the safety constraint is met for all models in the ambiguity set.

The authors propose an algorithm based on the Lagrange multiplier, which transforms the policy optimization problem into a min-max optimization problem. For such a problem, a gradient descent-ascent algorithm is proposed. The authors also establish sample complexity results about how many samples are needed to ensure convergence to an approximate stationary point that also approximately satisfies the safety constraints.



**Summary Of The Review:**

The main issue of this paper seems to be a lack of novelty. The algorithm and theoretical results seems to be based on direct combination of results in CMDP and robust MDP literature.

---

### Official Review · Reviewer_TPvy · 2022-10-24

**Confidence:** 4
**Correctness:** 2
**Technical Novelty And Significance:** 3
**Empirical Novelty And Significance:** Not applicable
**Recommendation:** 5

**Clarity, Quality, Novelty And Reproducibility:**

- Citation format should be cleaned up. citep and citet are not used properly. This has made the paper less readable.
- $G_t$ are re-used in both (9) and (17). Please change one.

**Strength And Weaknesses:**

Strength:
+ This work studies an important and timely problem of robust constrained RL.
+ The proposed robust primal-dual algorithm is valuable.
+ The theoretical analysis is technically sound (under the given assumptions and the chosen approach).

Weakness:
- The problem formulation has some inconsistency in terms of the (uncertain) MDP. The formulation in eq. (3) will mostly result in different worst-case MDPs for the "worst-case reward" and  the "worst-case cost". However, the motivating example is that the training MDP is different than the test MDP. In fact, majority, if not all, of practical applications should have the same MDP where both reward and cost are evaluated, as they are simultaneously collected. This should not be over different MDPs. The authors have specifically discussed the worst-case transition kernels being possibly different for the two robust value functions, but I am having a difficult time understanding why this is useful to begin with.
- The authors essentially got around the non-differentiable robust Bellman operator issue by taking a differentiable approximation. I understand that such approximation is made in order to make progress. However, despite that this approximation can be close to the original function, any approximation in a **robust** optimization formulation always raises the question of how these two ``uncertainties'' reconcile. Would solving the approximated problem lead to actual constraint violation? How does this affect the rigor of the original problem? Additionally, one can imagine that there could be different ways to do this approximation --  why did the authors choose the current approximation, and how is this superior than others?
- The above issue also happens when the authors propose to *estimate* the smoothed robust functions. Again, how does the estimation error affect the constraint violation and the rigor of the optimization problem?
- Assumption 2 and Lemma 4 have two lower bound values $\zeta$ and $\zeta'$. Are these two values needed for Alg. 1 and eq. (16), both of which need $\Lambda^*$? How can we have these values in practice? Knowing $\Lambda^*$ exists (finite) and selecting its value are non-trivial.

**Summary Of The Paper:**

This paper studies the problem of robust constrained reinforcement learning (RL) where the underlying model has uncertainty. The goal is to ensure that the constraints are satisfied for the worst-case MDP in a set of MDPs, while at the same time to maximize the reward over the uncertainty set. A robust primal-dual algorithm is proposed, with theoretical performance analysis under certain assumptions. An example of $\delta$-contamination uncertainty set is specifically investigated. Three numerical examples are shown to support the validity of the proposed solution.

**Summary Of The Review:**

Overall, this work made progress in the robust constrained RL problem, which is important and timely. The proposed algorithm has its value and the analysis is sound, but there are quite a few issues that are not resolved. This leads to my recommendation of marginal rejection.

---

### Official Review · Reviewer_gZnw · 2022-10-30

**Confidence:** 3
**Correctness:** 4
**Technical Novelty And Significance:** 3
**Empirical Novelty And Significance:** 2
**Recommendation:** 6

**Clarity, Quality, Novelty And Reproducibility:**

The paper itself is fairly clear and would benefit from some additional plain-english names for the quantities such as the robust estimation of rewards / costs. Overall, the paper is easy to follow and the underlying ideas are straightforward once the variables are understood as the paper itself is well-written.
Additionally, the authors would benefit from some examples of where this model might be used where the domain-expert wants to train a model that robustly gives high worst-case performance and satisfies constraints in the worst-case. Even though it seems clear that practitioners would want this, a couple of working examples would help make this more concrete.
The domain itself seems somewhat novel given the referenced related work, although I don’t work in the space of constrained RL. However, it would be helpful to explain what differentiates the approach from constrained RL where the robustness requirement might simply be encoded into the costs.
To enable reproducibility, it would be helpful to either release code or further specify what models were used in the application domains.

Small comments:
P5: change to “converges more stably and faster”


**Strength And Weaknesses:**

The main strengths of the paper are in the theoretical evaluation of their algorithm in their specific setting and in evaluating their algorithm against a reasonable baseline showing improved performance. Additionally, their algorithm is reasonable and straightforward and they evaluate the complexity of their convergence.

The main weaknesses of the paper are in the experimental evaluation. It is unclear whether the results demonstrate that the learned model satisfies the constraint robustly as is desired, although it clearly satisfies the constraints “better” than previous heuristic approach and a standard approach. Additionally, it would be helpful to explain exactly what metrics are used since it is slightly unclear whether the evaluation is a worst-case evaluation of the incurred costs and rewards, or just the incurred costs and rewards. Additionally, it is unclear whether the worst-case evaluations come from the approximation or are evaluated by finding the worst-case scenario from the uncertainty set.


**Summary Of The Paper:**

The authors present a method for robust constrained RL, a setting where the RL agent is tasked with maximizing the worst-case expected reward subject to a constraint that worst-case costs incurred from the environment are constrained. Here the worst-cases are evaluated as the worst-case among models of the environment in some uncertainty set. The algorithm itself optimizes the lagrangian of the constrained problem by performing gradient updates. To evaluate the expected worst-case reward and constraint, the authors leverage previous work that computes smoothed robust value estimates using log sum exp as an approximation for the worst-case setting. The authors demonstrate that this algorithm should achieve a solution with bounded infeasibility with guarantees on convergence and runtime due to assumptions on the problem. The authors evaluate their algorithm against a constraint-agnosting RL agent as well as a heuristic algorithm for robust constrained reinforcement learning from previous work.


**Summary Of The Review:**

Overall, I think that the paper approaches an important problem for the RL community. I see the main strength of the paper as presenting a theoretically-motivated approach to solving their problem and evaluating the approach against a reasonable baseline on several simulated domains. I consider the main room for improvement for this work to be in the experimental evaluation and expanding upon whether or not the approach meets the proposed robustness requirements.

---

### Decision · Program_Chairs · 2023-01-20

**Decision:**

Reject

**Justification For Why Not Higher Score:**

There are possibly technical flaws in the theoretical analysis. I would only accept a theoretical paper where at least correctness is not contested.


**Justification For Why Not Lower Score:**

N/A

**Metareview: Summary, Strengths And Weaknesses:**

The paper proposes an (at first sight) simple combination of robust and constrained MDPs. Indeed, solving a robust MDP problem can be cast a solving a CMDP problem. However, the gradient-ascent-descent method used is not very straightforward and requires the use of various approximation. There was a technical discussion which, at some point, was broken off by the reviewer.

I would rather err on the side of caution, defer to the reviewer, and reject this paper.